# Finding New Molecular Targets of Two Copper(II)-Hydrazone Complexes on Triple-Negative Breast Cancer Cells Using Mass-Spectrometry-Based Quantitative Proteomics

**DOI:** 10.3390/ijms24087531

**Published:** 2023-04-19

**Authors:** Lucia M. Balsa, María R. Rodriguez, Verónica Ferraresi-Curotto, Beatriz S. Parajón-Costa, Ana C. Gonzalez-Baró, Ignacio E. León

**Affiliations:** 1CEQUINOR (UNLP, CCT-CONICET La Plata, Asociado a CIC), Departamento de Química, Facultad de Ciencias Exactas, Universidad Nacional de La Plata, La Plata 1900, Argentina; 2Instituto de Física La Plata, IFLP (UNLP, CCT-CONICET La Plata), Departamento de Física, Facultad de Ciencias Exactas, Universidad Nacional de La Plata, La Plata 1900, Argentina; 3Cátedra de Fisiopatología, Departamento de Ciencias Biológicas, Facultad de Ciencias Exactas, Universidad Nacional de La Plata, La Plata 1900, Argentina

**Keywords:** breast cancer, molecular targets, metallodrugs, copper(II), proteomics

## Abstract

Breast cancer is the most common cancer in women, with a high incidence estimated to reach 2.3 million by 2030. Triple-Negative Breast Cancer (TNBC) is the greatest invasive class of breast cancer with a poor prognosis, due to the side-effects exerted by the chemotherapy used and the low effectivity of novel treatments. In this sense, copper compounds have shown to be potentially effective as antitumor agents, attracting increasing interest as alternatives to the usually employed platinum-derived drugs. Therefore, the aim of this work is to identify differentially expressed proteins in MDA-MB-231 cells exposed to two copper(II)-hydrazone complexes using label-free quantitative proteomics and functional bioinformatics strategies to identify the molecular mechanisms through which these copper complexes exert their antitumoral effect in TNBC cells. Both copper complexes increased proteins involved in endoplasmic reticulum stress and unfolded protein response, as well as the downregulation of proteins related to DNA replication and repair. One of the most relevant anticancer mechanisms of action found for CuHL1 and CuHL2 was the down-regulation of gain-of-function-mutant p53. Moreover, we found a novel and interesting effect for a copper metallodrug, which was the down-regulation of proteins related to lipid synthesis and metabolism that could lead to a beneficial decrease in lipid levels.

## 1. Introduction

Cancer is one of the main causes of death worldwide [1]. Breast cancer is the most common cancer in women worldwide and one of the more frequent causes of premature mortality in the female population [2]. There are two types of breast cancer, ductal and lobular, which are divided into invasive and in situ (non-invasive) types, with several subtypes based on histology features. One of the most aggressive classes of breast cancer is the Triple-Negative Breast Cancer (TNBC), which does not express estrogen, progesterone, and HER-2 receptors [3]. For TNBC, the treatment options are limited to chemotherapy, including anthracyclines (doxorubicin and epirubicin), taxanes (paclitaxel and docetaxel), and capecitabine, as hormone or targeted therapy cannot be utilized [4]. Platinum chemotherapeutics such as carboplatin are also used [5]. Nevertheless, these treatments present a low efficacy as they produce important adverse effects and a high rate of metastatic recurrence [6]. Therefore, great efforts are dedicated to developing new strategies using therapeutic agents to improve and optimize the treatment. To this end, several metal-based drugs including palladium [7], ruthenium [8,9], and copper [10] were designed and many of them displayed antitumor activity on breast cancer cells.

In this sense, copper compounds offer a promising and innovative alternative to breast cancer treatment. Interest in them arises from the knowledge that copper can play an important role as a limiting factor in different aspects of tumor progression, such as growth, angiogenesis, and metastasis [11]. Several copper compounds showed promising antitumor and anti-metastatic properties on diverse kinds of solid tumors [12,13,14]. The main mechanisms of action reported involve reactive oxygen species (ROS) generation, glutation (GSH) depletion, proteasome inhibition, and DNA damage [15,16,17,18,19]. However, details of the cell signaling and molecular mechanism of copper complexes are mainly still unanswered.

Proteomics is a very powerful research tool for cellular processes as it provides detailed information of the fine alterations in cell homeostasis triggered by exposure to drugs. Alterations in intracellular signaling and metabolic pathways may tell us which parts of the cellular machinery are mainly affected by treatment, thus suggesting the most probable biomolecular targets for the compound [20].

We have previously reported the synthesis, physicochemical characterization, and antitumoral activity of two copper(II)-hydrazone complexes, [Cu(HL1)(H_2_O)](NO_3_).H_2_O and [Cu(HL2)(H_2_O)_2_](NO_3_), or CuHL1 [21,22] and CuHL2 [23] for simplicity. In vitro studies revealed a promising anticancer activity of both complexes against breast cancer cell lines, including the TNBC line MDA-MB-231. Moreover, some possible action mechanisms were demonstrated, such as DNA damage, ROS production, and proteasome inhibition [22,23]. However, the key signal pathways underlying the anticancer mechanism and the therapeutic targets of CuHL1 and CuHL2 have not yet been well characterized. Therefore, this study was initiated to identify differentially expressed proteins in MDA-MB-231 cells exposed to CuHL1 and CuHL2 treatment, using label-free quantitative proteomics and functional bioinformatics strategies to identify the molecular mechanisms through which these copper complexes exert their antitumoral effect in TNBC cells.

## 2. Results and Discussion

### 2.1. Synthesis and Characterization of the Copper Complexes

The compounds CuHL1 and CuHL2 (Figure 1) were obtained following the procedure described in our previous works. They crystallize as complex cations with a +1 charge and with nitrate as a counterion [21,23].

The corresponding hydrazone (H_2_L1: 2-acetylthiophene-2-hydroxy-3-methoxybenzohydrazone and H_2_L2: 2-acetyl-4-methoxyphenyl-2-hydroxy-3-methoxybenzohydrazone) coordinates to the metal as a monoanionic ligand (HL^−^), by deprotonation of the phenolic oxygen, through the ONO chelating system. The coordination sphere of CuHL1 and CuHL2 is completed with one and two water molecules, respectively, according to the previously reported crystallographic results.

In previous works, we have demonstrated their antitumoral activities against breast cancer cells. Both complexes impaired cell viability in the sub-micromolar concentration range (0.5–2 µM) against breast cancer cell lines. Particularly, the IC_50_ values in the TNBC cell line MDA-MB-231 are 1.6 ± 0.2 µM (CuHL1) and 1.6 ± 0.1 µM (CuHL2) [22,23].

### 2.2. Label-Free Mass Spectroscopy Quantification of Proteins Isolated from MDA-MB-231 Cells following Treatment with CuHL1 and CuHL2

To explore the anticancer mechanism of CuHL1 and CuHL2, we conducted label-free quantitative proteomics profiling.

Label-free quantification using the Orbitrap LC–MS/MS (Thermo Scientific^TM^, Waltham, MA, USA) was able to identify proteins that are differentially expressed between cells treated with CuHL1 or CuHL2 and untreated conditions. A total of 1656 and 1659 proteins were identified for CuHL1- and CuHL2-treated cells, respectively. The resulting proteomic dataset was filtered by fold-change differences and a significant *p*-value (Figure 2). From the CuHL1-treated cells, a total of 69 proteins were identified to be differentially expressed when compared with the untreated control (Table 1). Among these differentially expressed proteins, 28 proteins were up-regulated and 41 proteins were down-regulated. As from the CuHL2-treated cells, a total of 63 proteins were identified to be differentially expressed when compared with the basal condition (Table 2). Among these differentially expressed proteins, 23 were up-regulated and 40 were down-regulated by the treatment.

### 2.3. Functional GO Enrichment Analysis

To better understand the biological characteristics of the differentially expressed proteins, a set of bioinformatics tools were applied. First, the STRING enrichment analysis was used to study whether any Gene Ontology (GO) categories were statistically enriched. For each complex, two different lists containing the up-regulated and down-regulated proteins were uploaded. The GO database classifies functions in organisms into three categories: the involved biological process, the molecular function, and the cell component. Figure 3 displays the obtained GO terms statistically over-represented (*p*-value ≤ 0.05 after Benjamini correction). CuHL1 did not show enriched terms of molecular functions for down-regulated proteins, while CuHL2 did not show enriched terms of the cell component for up-regulated proteins.

Among up-regulated proteins, the principal biological processes enriched for both complexes were related to Heat Shock Proteins (HSPs) and co-chaperones: “Response to unfolded protein”, “Response to heat”, and “Chaperone cofactor-dependent protein refolding”. The proteins involved in these processes are HSPs and co-chaperones, mainly from the HSP70 family (HSPA1B, HSPH1, HSPA6, BAG3, DNAJA1, and DNAJB1). In terms of molecular function, mostly proteins assigned to the binding activity associated with HSPs and co-factors could be found for CuHL1 and CuHL2. 

An increase in HSP expression levels upon treatment with anticancer agents has been reported. These proteins may participate in the stress response to drug-induced damage [24]. HSPs are involved in protein folding and are generally expressed as a reaction to endoplasmic reticulum stress. Cancer cells have enhanced ER stress due to several characteristics, such as hypoxia, low nutrient availability, lactic acidosis, oxidative stress, and increased replication and metabolism, which leads to increased protein folding and accumulation of misfolded proteins. In response to ER stress, the Unfolded Protein Response (UPR) is initiated [25]. The primary aim of UPR is to re-establish ER homeostasis by increasing the protein-folding capacity [26]. Cancer cells up-regulate the UPR pathway to increase their ability to survive under heightened ER stress. However, if the ER stress is prolonged or acute, a terminal UPR program promotes cell death. In this sense, many cancers with enhanced UPR can be hypersensitive to compounds that generate ER stress. Chemotherapeutics can achieve UPR disruption via different mechanisms such as inhibition of proteasome activity, oxidative stress, and alteration of ER Ca^2+^ storage. The clinically approved bortezomib and carfilzomib are proteasome inhibitors that interfere with protein degradation, enhancing ER stress. Moreover, ROS generation and proteasome inhibition are reported as the main mechanisms of action of copper complexes [13]. Tardito et al. reported a thioxotriazole copper(II) complex capable of inducing UPR via inhibition of the ubiquitin proteasome system. Gene expression profiling showed that the complex up-regulated genes related to the unfolded protein response. Functional analysis revealed an enrichment of chaperone and unfolded protein binding categories [27]. We have previously reported that CuHL1 inhibits proteasome activity [22] and that the proteomic analysis showed a down-regulation of proteasome subunit PSMD4. On the other hand, we have determined that CuHL2 induces a significant increment in the ROS levels [23]. These mechanisms could lead to ER stress and accumulation of unfolded proteins that could explain the up-regulation of HSPs with both complexes.

When we analyzed the down-regulated proteins, we found more differences in GO terms between the two treatments. This is due to a lower proportion of common proteins between CuHL1 and CuHL2 among down-regulated proteins. Shared proteins represent 44% and 45% of down-regulated proteins for CuHL1 and CuHL2, respectively. Meanwhile, the shared up-regulated proteins represent 50% and 61% for CuHL1 and CuHL2, respectively. The only biological process in common is the “Organic substance metabolic process”. After CuHL1 treatment, down-regulated proteins were enhanced in other metabolic processes, such as the “Small-molecule metabolic process”, “Oxidation–reduction process”, and “Carboxylic acid metabolic process”. 

On the other hand, CuHL2 presents biological processes related to DNA replication and repair. This difference is replicated when we analyzed the Cellular Component. Proteins down-regulated with CuHL2 are found in the “Chromosome telomeric region”, “CMG complex”, and “MCM complex”, which are components related to DNA replication. In a previous work, we demonstrated that CuHL2 is capable of interacting with DNA and cause damage [24]. This could be the reason for the inhibition of proteins related to DNA replication.

### 2.4. Protein–Protein Interaction Analysis

The STRING v11.5 database was used to perform Protein–protein interaction (PPI) analysis to evaluate the interactions between the differentially expressed proteins in response to treatment for both copper complexes. In these networks, the nodes represent proteins, and the edges represent the interactions between the two proteins, whereby the line thickness between two nodes indicates the strength of data support. For all PPI networks, Markov Cluster Algorithm (MCL) clustering was performed to identify significant protein–protein interaction clusters formed among differentially expressed proteins.

Figure 4 displays the networks generated for up-regulated proteins after treatment with CuHL1 and CuHL2. In both networks, prominent interaction clusters were generated containing HSPs and co-chaperones proteins: BAG3, CHORDC1, DNAJA1, DNAJB1, HSPA1B, and HSPH1 for CuHL1; AHSA1, BAG3, CHORDC1, DNAJA1, DNAJB1, HSPA1B, HSPA6, and HSPH1 for CuHL2. These clusters correlate with the enriched biological processes found in GO analysis.

Both down-regulated networks (Figure 5) display TP53 as a central node. The TP53 tumor suppressor gene encodes the p53 protein, a transcription factor that is crucial for proper control of cell cycle progression, senescence, apoptosis, DNA repair, and genome maintenance among other important functions [28]. The TP53 gene is the most frequent target for mutation in tumors, mutated in over half of all cancers [29]. Moreover, TP53 mutations are highly frequent and one of the key driving factors in triple-negative breast cancer [30]. There are TP53 mutations that result in a “loss of function” that eliminates the tumor suppressing effects of p53. However, many mutations to TP53 are “gain-of-function” (GOF) mutations that can acquire oncogenic properties, augmented invasiveness, metastasis, and recurrence of cancer [31,32]. The expression of mutant p53 in preclinical breast cancer models showed a correlation with increased survival, migration, invasion, and metastasis [30,32,33]. Most mutant p53s are expressed at very high levels in cancer cells, so the degradation or inhibition of their activity can be considered promising therapeutic mechanisms [34]. Particularly, MDA-MB-231 cells present a highly expressed GOF mutant p53. It has been demonstrated that a reduction of mutant p53 can induce apoptosis in MDA-MB-231 cells [35]. Therefore, the down-regulation of mutant p53 is one of the most important antitumor mechanisms of action found for CuHL1 and CuHL2.

Another similarity in both down-regulated networks is the presence of clusters related to lipid metabolism. Figure 4b presents a cluster form with CYP51A1, FADS2, and LDLR; while the cluster in Figure 3d includes proteins ACOT2, ACSL4, CYP51A1, FADS2, FASN, LDLR, and PCK2. Altered lipid metabolism is a recognized factor in cancer metabolism. Tumor cells tend to increase the novo lipogenesis, lipid uptake, and storage, which leads to an increment in source material for the biogenesis of cell membranes, as well as in energy supplies via β-oxidation of fatty acids, and in an increase in the lipid signaling molecules that mediate oncogenic pathways [36]. This lipid metabolism dysregulation helps to promote tumor growth, metastatic spread, and therapy resistance [37]. The up-regulation of lipogenic enzymes has been reported in several cancers including breast, prostate, colorectal, ovarian, gastrointestinal, and lung cancer [38,39]. Consequently, targeting-altered lipid metabolism pathways have been studied as a promising anticancer therapy.

One of the most down-regulated proteins after treatment with CuHL1 was Lanosterol 14-alpha demethylase (CYP51A1). This protein catalyzes one of the key steps in cholesterol biosynthesis and is usually overexpressed in tumor cells [40]. In fact, Kerber et al. demonstrated that de novo cholesterol synthesis was blocked when CYP51A1 was knocked-out [41]. Furthermore, the presence of a lanosterol 14-alpha demethylase inhibitor led to the induction of apoptosis in cancer cells [42]. In this sense, several pre-clinical and clinical studies have focused on targeting cholesterol metabolism as a treatment for various cancers [43]. The strategies include aiming for cholesterol biosynthesis pathways, as well as the exogenous-sterol uptake. For example, current clinical trials have demonstrated the protective effect in breast cancer of drugs that inhibit the mevalonate pathway such as statins and Zoledronate [44].

On the other hand, many cancer cells exhibit an overexpression of LDLR, which facilitates the rapid uptake of LDL cholesterol and contributes to the accumulation of lipid components [45,46]. Thus, the downregulation or inhibition of LDLR affects cholesterol uptake and could increase the efficacy of chemotherapeutic drugs. Guillaumond et al. demonstrated that the shRNA silencing of LDLR reduced the cholesterol uptake and diminished the proliferation of pancreatic cancer cells [47]. For breast cancer, Gallagher et al. showed that elevated LDLR expression and high LDL levels are significant for tumor growth and that silencing LDLR in TNBC cells increases cell death and reduces the growth of tumors [48]. Proteomics showed that LDLR was down-regulated after treatment with both complexes in MDA-MB-231 cells. This TNBC cell line presents higher expression levels of LDLR compared to the estrogen-receptor-positive MCF7, or the non-tumorigenic MCF-10A cell lines [48].

Other proteins related to lipid metabolism were down-regulated by CuHL1 and CuHL2. They are key enzymes that participate in fatty acid metabolism, such as fatty acid synthase (FASN), fatty acid desaturase (FADS2), and long-chain fatty acid synthase (ACSL4). Fatty acid synthesis and palmitoleic acid generation are enhanced and play an essential role in cancer growth [36]. FASN and ACSL4 are highly expressed in many cancers including breast cancer [49,50], thus making them an attractive target for inhibiting cancer cell proliferation. In this sense, Cui et al. showed that inhibiting FAS by inhibitors or shRNAs induces apoptosis in breast cancer cells [51]. Another FASN inhibitor, Fasnall, was able to inhibit breast cancer cell growth, induce apoptosis, and showed potent in vivo antitumor activity against breast cancer, alone and combined with carboplatin [52]. Similarly, ACSL4 knockdown inhibited the cell proliferation of several cancer cell lines [53,54]. In breast cancer, it was demonstrated that ACSL4 targeting increased the efficacy of Triacsin C, as combined treatment with ACSL4 inhibition resulted in synergistic antitumor effects [55]. Overall, the down-regulation of these proteins by CuHL1 and CuHL2 could lead to a beneficial decrease in lipid synthesis and metabolism, which is a novel and interesting effect for a copper metallodrug.

CuHL1 presented a cluster in the down-regulated network that includes proteins also related to cell metabolism, specifically associated with the tricarboxylic acid (TCA) cycle and pyruvate metabolism. The cluster forms with all mitochondrial proteins: ACO1, ALDH2, DLAT, IDH3A, PC, PCK2, and TST. This cluster correlates with the enriched metabolic processes in GO functional analysis: “Small-molecule metabolic process”, “Oxidation–reduction process”, and “Carboxylic acid metabolic process”. CuHL1 could generate the inhibition of mitochondrial metabolism. Traditionally, “the Warburg effect” hypothesizes that most cancer cells rely on aerobic glycolysis to engender the energy needed for cellular activity, rather than mitochondrial oxidative phosphorylation [56]. However, recent studies indicate that active mitochondrial metabolism is necessary for tumor growth as it provides key metabolites for macromolecule synthesis and generates oncometabolites to sustain the phenotype of cancer cells [57]. In particular, recent studies have shown that TNBCs have an altered metabolic profile, characterized by the elevated uptake and utilization of glucose, glutamine, and TCA cycle intermediates in addition to increased fatty acid β-oxidation [58,59,60]. As a consequence, the down-regulating effect of CuHL1 on several proteins related to mitochondrial metabolism could be beneficial for breast cancer therapy.

Moreover, Tsvetkov et al. recently defined a new copper-dependent form of regulated cell death related to mitochondrial metabolism called cuproptosis. In this cell-death mechanism, excess copper binds to lipoylated enzymes of the TCA cycle, resulting in lipoylated protein aggregation that leads to proteotoxic stress and ultimately cell death [61]. One of the lipoylated proteins that can bind to copper is DLAT, which is down-regulated after treatment with CuHL1. The decreased expression of DLAT could be the result of protein oligomerization caused by copper binding. In fact, Zhou et al. reported a copper nanoplatform that induces cuproptosis and generates the same down-regulating effect in DLAT [62]. Furthermore, the proteotoxic stress caused by protein oligomerization leads to the induction of HSP70, whose expression is increased by both complexes. This could indicate that CuHL1 is able to promote cuproptosis as a cell-death mechanism.

Another important cluster in the CuHL2 down-regulated network includes proteins that participate in DNA replication: MCM2, MCM3, MCM5, RRM2, and SMC2. These proteins are responsible for the related GO terms that were mentioned before. Nevertheless, in this analysis, we found that CuHL1 also has a cluster formed with proteins that participate in DNA replication: MCM5, POLD1, and POLDIP2. We have demonstrated, in in silico and in vitro studies, that both complexes were able to interact with DNA and produce damage in the macromolecule [22,23]. The down-regulation of minichromosome maintenance (MCM) proteins is present in both CuHL1 and CuHL2 treatments. The MCM protein family plays a key role in eukaryotic DNA replication. The MCM complex is a DNA replication licensing factor that controls the once-per-cell cycle DNA replication [63]. Dysregulation of the MCM complex has been associated with the occurrence and progression of many tumors [64]. Moreover, overexpression of MCM has been detected in various cancer cells, including breast cancer [65]. Additionally, CuHL1 presented a down-regulation of POLD1. The DNA polymerase delta (POLD) family is involved in DNA replication and is an important mediator of DNA repair during chromosome replication [66]. Mutation in POLD can be associated with cancer development and it was demonstrated that POLD1 is able to affect cell cycle progression and promote cancer cell proliferation [67]. In breast cancer, gene and protein expression levels of POLD1 are elevated [68]. Moreover, survival analysis demonstrated an association of increased POLD1 levels with poor disease-free survival, late-stage cases, and the presence of TNBC [69]. In this sense, several studies report that the down-regulation of POLD1 in breast cancer cells suppressed cell cycle progression and cell proliferation and promoted apoptosis [69,70].

In order to analyze the shared mechanisms of CuHL1 and CuHL2, we performed a PPI network analysis utilizing only the common proteins between both treatments. 

Figure 6a displays the network generated for the shared up-regulated proteins. The interaction cluster generated includes HSPs and co-chaperone proteins: BAG3, CHORDC1, DNAJA1, DNAJB1, HSPA1B, and HSPH1. This cluster demonstrates the shared mechanism of ER stress and UPR induction for both complexes.

The down-regulated network (Figure 6b) presents the p53 central node. Moreover, a cluster related to lipid and cholesterol metabolism is shown. The cluster includes the proteins: LDLR, CYP51A1, and FADS2. This way, other shared mechanisms between CuHL1 and CuHL2 include the down-regulation of GOF-p53 and proteins involved in lipid metabolism.

### 2.5. Ingenuity Pathway Analysis

Finally, the ingenuity pathway analysis (IPA) bioinformatics tool was run to evaluate the most affected canonical pathways after treatment with both complexes. The over-represented canonical pathways are reported in Table 3 and Table 4.

In correlation with the results of up-regulated proteins analysis, both complexes displayed an alteration in stress-response pathways such as “Unfolded protein response”, “NRF2-mediated oxidative stress response” [71], and “Autophagy”. The latter can be positively stimulated by the UPR program because, under ER stress conditions, ER produces several signals that stimulate autophagy [72]. Moreover, CuHL1 presented an enrichment of the “Protein ubiquitination pathway” and “EIF2 signaling”, which is involved in one of the UPR pathways [25].

Molecular pathways related to cell death were enhanced. Both complexes displayed alterations in “Induction of Apoptosis by HIV1”. “Apoptosis signaling” was also altered by CuHL1. Meanwhile, CuHL2 displayed an enhancement of “Immunogenic cell death” (ICD). ICD is a form of cell death where the dying cancer cells stimulate immune cells to actively seek and destroy them. ICD is characterized by the emission of a class of the danger-associated molecular patterns (DAMPs) family, which functions as “find me” and “eat me” signals to the tumor-associated immune cells. The key DAMPs include the release of adenosine triphosphate (ATP), high-mobility group protein B1 (HMBG1), and exposed molecules on the outer membrane such as CRT (CRT) and heat-shock proteins (Hsp90 and Hsp70) [73]. In this sense, both complexes have Hsp70 proteins as the most up-regulated protein. Moreover, the ability of cancer therapies to induce ICD depends on their ability to induce ER stress and ROS production. ROS-based ER stress is an essential component to trigger DAMPs expression and the intracellular danger signaling pathways [74]. Redox stress induction is the principal mode of action for several anticancer copper complexes. Kaur et al. reported a Cu(II) complex with a Schiff base ligand that is capable of inducing ICD in breast cancer stem cells through elevation of ROS levels and induction of ER stress [75].

Angiogenesis was another process affected by both complexes, as seen by the alteration of “Inhibition of Angiogenesis by TSP1” and “HIFa signaling”. In this sense, CuHL1 presented a cluster in the down-regulated network (Figure 4b) related to vascular endothelial growth factor (VEGF) signaling and angiogenesis, formed with proteins ENG, GIPC1, IGFBP7, and SDC4. Angiogenesis is essential for breast cancer progression and dissemination. Several molecular pathways are known to drive angiogenic switches in cancer cells [76]. Hence, the inhibition of pro-angiogenic pathways is a promising therapeutic alternative. There are numerous clinical and pre-clinical studies on targeting angiogenic pathways in breast cancer [77].

Finally, DNA replication and repair pathways were also affected, as seen in the GO functional analysis and PPI networks. Both complexes displayed an alteration of “Cell cycle control of chromosomal replication” and “Ribonucleotide reductase (RR) signaling pathway”. RR catalyzes the reduction of ribonucleotides to their corresponding deoxyribonucleotides, so RR is essential for DNA replication and repair [78]. The expression of subunit RRM2 is dysregulated in multiple cancer types, including breast cancer. Particularly, the MDA-MB-231 cell line presents an increased expression of RRM2 [79].

## 3. Materials and Methods

### 3.1. Synthesis, Identification, and Preparation of CuHL1 and CuHL2

Both copper(II) compounds were obtained following the procedure defined in our previous manuscripts [21,23].

Fresh stock 20 mM solutions of the complexes were prepared in dimethylsulfoxide (DMSO) and forward-diluted according to the concentrations used in each experiment. The maximum concentration of DMSO was maintained at 0.5% to avoid the toxic effects of this solvent on the cells.

### 3.2. Cell Culture

MDA-MB-231 breast cancer cells were grown in Dulbecco’s modified Eagle’s medium Nutrient Mixture F12 (DMEM F12) with 10% fetal bovine serum (FBS), 100 IU/mL of penicillin, and 100 μg/mL of streptomycin at 37 °C in a 5% CO_2_ atmosphere.

### 3.3. Protein Sample Preparation

For sample preparation, MDA-MB-231 cells were seeded in a 6-well dish, allowed to attach for 24 h, and treated with 1 µM of the complex at 37 °C. Triplicates of each condition were used. Total protein was extracted from MG-63 cells after 24 h. Briefly, cells were homogenized in RIPA lysis buffer containing a protease inhibitor cocktail. Then, total protein was collected through centrifugation at 12,000× *g* for 20 min at 4 °C, and protein concentration was determined using the BCA protein assay.

### 3.4. Protein Identification and Mass Spectrometry

Samples were sent to the Center for Chemical and Biological Studies by Mass Spectrometry (CEQUIBIEM) for Label-Free Quantification analysis. In brief, samples were reduced with DTT, alkylated with iodoacetamide, and followed by trypsin digestion. Samples were lyophilized by Speed Vac and resuspended in 0.1% trifluoroacetic acid. Then, liquid chromatography was performed with the nanoHPLC Easy nLC 1000 (Thermo Scientific, Waltham, MA, USA) coupled to a mass spectrometer with Orbitrap technology (Thermo Scientific, Waltham, MA, USA), which allows separation and further identification of the peptides.

Analysis of the spectra obtained by the mass spectrometer was performed using the Proteome Discoverer search engine with the Homo sapiens database. For the search, the following parameters were set: trypsin was used as the cleavage protease; two missed cleavages were allowed; the precursor peptide mass tolerance was set at 10 ppm while the fragment mass tolerance was 0.05 Da; carbamidomethylation (C) was set as a fixed modification; the variable modification was set to oxidation; the minimum identification criteria required a minimum of 2 peptides per protein.

Statistical analysis for differentially expressed proteins was performed using the software Perseus v.1.6.6.0. The t-test was used to compare protein abundance averages between treatment and control groups. Differentially expressed proteins were identified when the *t*-test *p* value < 0.05 and there was an increase or decrease in the protein level of 2-fold or more.

### 3.5. Bioinformatics Analysis

#### 3.5.1. Functional GO Enrichment Analysis

The differentially expressed proteins were used to perform Gene Ontology (GO) enrichment analysis with the Search Tool for the Retrieval of Interacting Genes (STRING) enrichment API method (https://string-db.org, accessed on 12 December 2022). For that, the up- and down-regulated protein IDs were submitted for Homo sapiens and were categorized according to their molecular function, biological process, and cellular component. For each of the different GO categories, the False Discovery Rate and Bonferroni-corrected *p*-values were calculated.

#### 3.5.2. Protein–Protein Interaction Analysis

The STRING v11.5 database was used to predict functional interactions between differentially expressed proteins and to map the protein–protein interaction networks (http://string-db.org, accessed on 12 December 2022). Protein IDs were submitted into the multiple protein analysis, and interaction sources were selected: text mining, experiments, databases, co-expression, neighborhood, gene fusion, and co-occurrence. A default medium confidence threshold (0.4) was used to define protein–protein interactions. MCL clustering (inflation parameter 3) was applied to the analysis to identify protein groups with similar interactions.

#### 3.5.3. Ingenuity Pathway Analysis

Pathway analysis was evaluated using Ingenuity Pathway Analysis (IPA). The dataset for each complex, combining up-regulated and down-regulated proteins, was uploaded into the QIAGEN IPA (QIAGEN Inc., Venlo, Netherlands) system for core analysis. IPA was performed to identify the most significant canonical pathways. The significance values (*p*-value) for the canonical pathways were calculated by the right-tailed Fisher’s Exact Test.

## 4. Conclusions

In this study, we conducted a label-free quantitative proteomic analysis of MDA-MB-231 cells treated with CuHL1 and CuHL2 to provide insight into the molecular mechanism of these copper complexes in TNBC cells. Bioinformatic and functional analysis revealed similar modes of action between both complexes. CuHL1 and CuHL2 treatment generated an increment of proteins involved in ER stress and UPR, as well as the down-regulation of proteins related to DNA replication and repair. One of the most important antitumor mechanisms of action found for CuHL1 and CuHL2 was the down-regulation of GOF-mutant p53. Additionally, we found a novel and interesting effect for a copper metallodrug, which was the down-regulation of proteins related to lipid synthesis and metabolism that could lead to a beneficial decrease in lipid levels. However, further functional studies are required to understand the specific mechanisms underlying the antitumoral effects of CuHL1 and CuHL2. It will be necessary to carry out experimental validations based on biochemical and other functional experiments, which go beyond the aim of the present work.

## Figures and Tables

**Figure 1 ijms-24-07531-f001:**
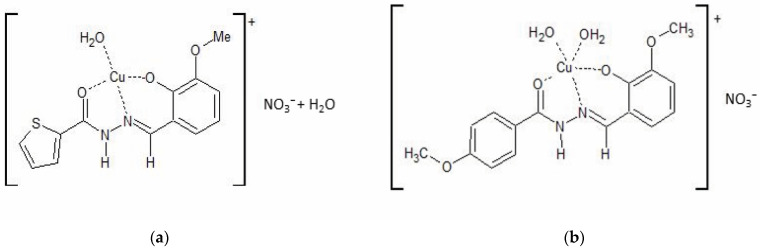
Schematic representation of (**a**) CuHL1 and (**b**) CuHL2.

**Figure 2 ijms-24-07531-f002:**
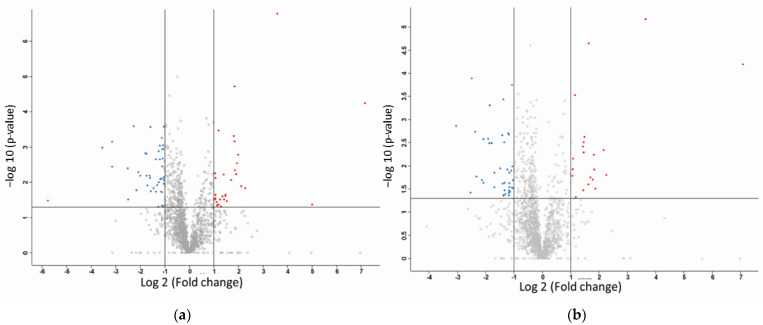
Volcano plots showing the variation in log10 (*p* value) with log2 (fold change) for various pairwise comparisons: (**a**) CuHL1 vs. Control and (**b**) CuHL2 vs. Control. Color map: grey: not significant, red: up-regulated, blue: down-regulated.

**Figure 3 ijms-24-07531-f003:**
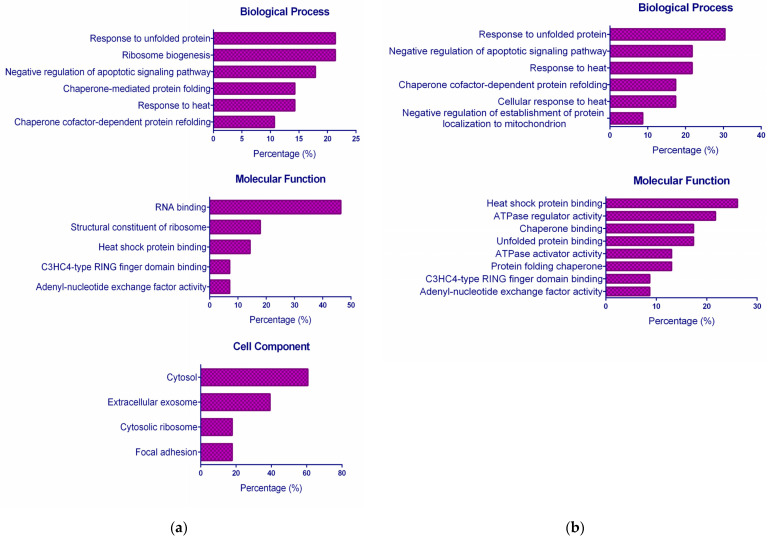
GO enrichment analysis of (**a**,**b**) up-regulated proteins after treatment with (**a**) CuHL1 and (**b**) CuHL2; (**c**,**d**) down-regulated proteins after treatment with (**c**) CuHL1 and (**d**) CuHL2. The percentage (%) of representation of each category was determined as a percentage of the total differentially expressed protein.

**Figure 4 ijms-24-07531-f004:**
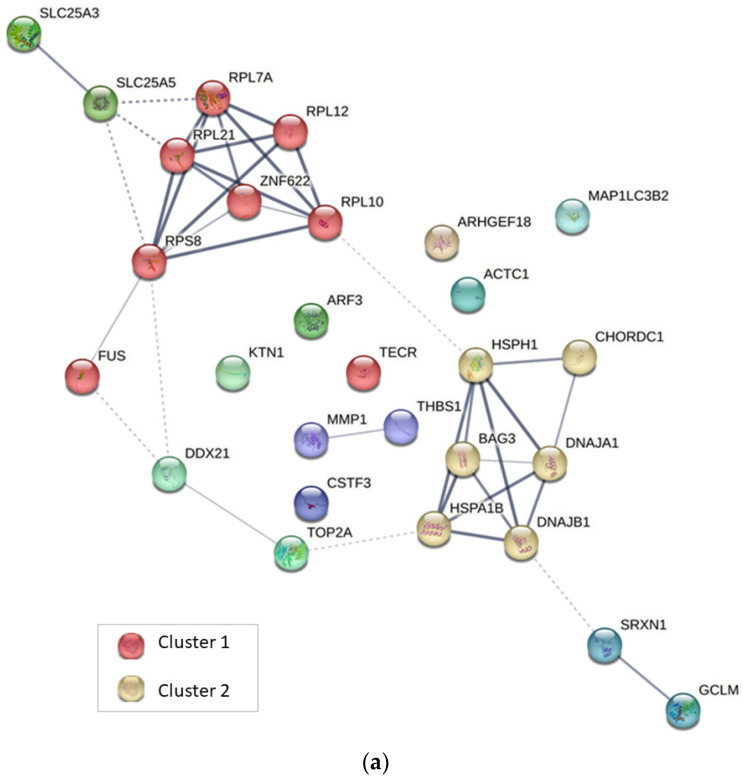
Interaction network for up-regulated proteins of MDA-MB-231 cells in response to treatment with (**a**) CuHL1 and (**b**) CuHL2 generated using the STRING v11.5 database. The thickness of the line indicates the degree of confidence of the interaction. A variety of interaction sources were included into the search strategy, such as text mining, experiment record, database record, coexpression, neighborhood, gene fusion, and co-occurrence.

**Figure 5 ijms-24-07531-f005:**
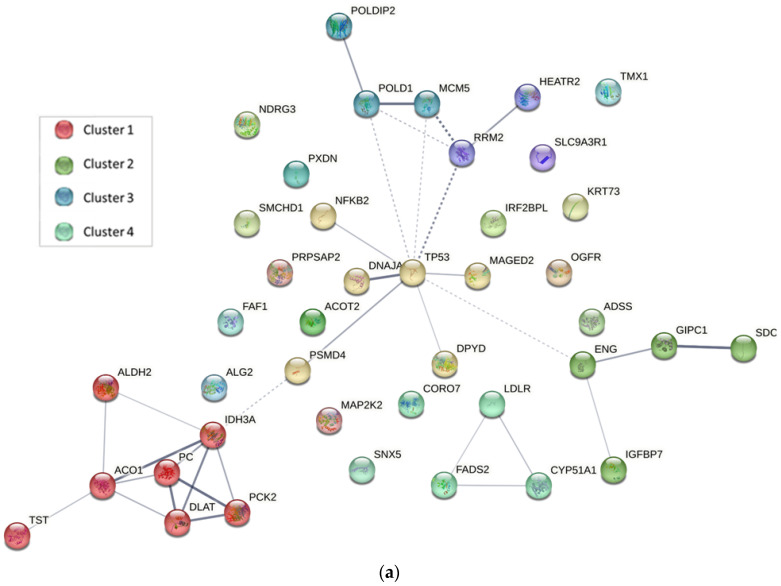
Interaction network for down-regulated proteins of MDA-MB-231 cells in response to treatment with (**a**) CuHL1 and (**b**) CuHL2 generated using the STRING v11.5 database. The thickness of the line indicates the degree of confidence of the interaction. A variety of interaction sources were included into the search strategy, such as text mining, experiment record, database record, coexpression, neighborhood, gene fusion, and co-occurrence.

**Figure 6 ijms-24-07531-f006:**
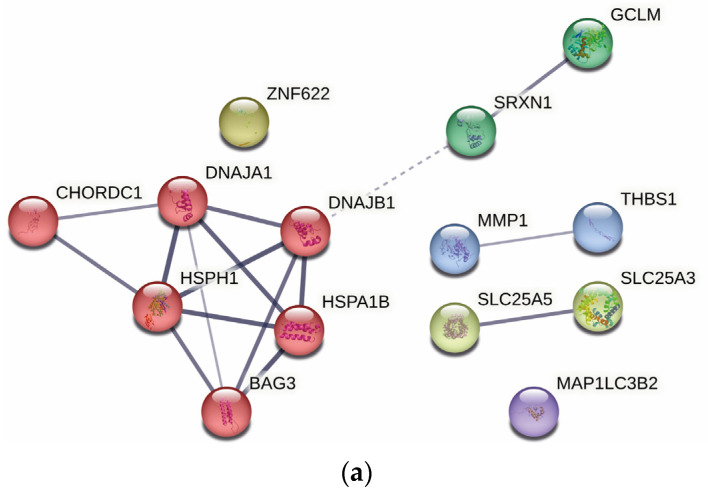
Interaction network for shared proteins between CuHL1 and CuHL2. (**a**) Up-regulated proteins; (**b**) Down-regulated proteins. Generated using the STRING v11.5 database. The thickness of the line indicates the degree of confidence of the interaction. A variety of interaction sources were included into the search strategy, such as text mining, experiment record, database record, coexpression, neighborhood, gene fusion, and co-occurrence.

**Table 1 ijms-24-07531-t001:** Differentially expressed proteins in CuHL1-treated MDA-MB.231 cells.

Gene	Protein	Fold-Change	*p*-Value
HSPA1B	Heat shock 70 kDa protein 1B	11.9067	0.0000
SRXN1	Sulfiredoxin-1	4.8384	0.0144
CSTF3	Cleavage stimulation factor subunit 3	4.3157	0.0128
THBS1	Thrombospondin-1	3.9412	0.0016
CHORDC1	Cysteine and histidine-rich domain-containing protein 1	3.8146	0.0028
BAG3	BAG family molecular chaperone regulator 3	3.7564	0.0058
SLC25A3	Phosphate carrier protein, mitochondrial	3.6397	0.0046
HSPH1	Heat shock protein 105 kDa	3.5744	0.0007
DNAJB1	DnaJ homolog subfamily B member 1	3.5735	0.0000
GCLM	Glutamate–cysteine ligase regulatory subunit	3.4930	0.0005
SLC25A5	ADP/ATP translocase 2	3.2590	0.0085
ARF3	ADP-ribosylation factor 3	2.8776	0.0335
MAP1LC3B2	Microtubule-associated proteins 1A/1B light chain 3 beta 2	2.7896	0.0222
DDX21	Nucleolar RNA helicase 2	2.7624	0.0244
KTN1	Kinectin	2.6451	0.0302
MMP1	Interstitial collagenase	2.6260	0.0060
ACTC1	Actin, alpha cardiac muscle 1	2.5367	0.0247
FUS	RNA-binding protein FUS	2.3718	0.0305
DNAJA1	DnaJ homolog subfamily A member 1	2.2815	0.0003
TECR	Very-long-chain enoyl-CoA reductase	2.2753	0.0422
ARHGEF18	Rho guanine nucleotide exchange factor 18	2.1868	0.0453
RPL12	60S ribosomal protein L12	2.1825	0.0442
ZNF622	Zinc finger protein 622	2.1629	0.0360
RPL21	60S ribosomal protein L21	2.0765	0.0283
TOP2A	DNA topoisomerase 2-alpha	2.0762	0.0075
RPS8	40S ribosomal protein S8	2.0729	0.0227
RPL7A	60S ribosomal protein L7a	2.0663	0.0055
RPL10	60S ribosomal protein L10	2.0453	0.0303
SDC4	Syndecan-4	−54.1800	0.0332
CYP51A1	Lanosterol 14-alpha demethylase	−11.7752	0.0010
FADS2	Fatty acid-desaturase	−8.9256	0.0007
ALG2	Alpha-1,3/1,6-mannosyltransferase ALG2	−8.8383	0.0036
RRM2	Ribonucleoside-diphosphate reductase subunit M2	−5.7327	0.0040
SMCHD1	Structural maintenance of chromosomes flexible hinge domain-containing protein 1	−5.6365	0.0302
DPYD	Dihydropyrimidine dehydrogenase [NADP(+)]	−4.8430	0.0003
POLD1	DNA polymerase	−4.4882	0.0165
NFKB2	Nuclear factor NF-kappa-B p100 subunit	−4.2297	0.0051
IRF2BPL	Probable E3 ubiquitin-protein ligase IRF2BPL	−3.9927	0.0064
TP53	Cellular tumor antigen p53	−3.4861	0.0015
LDLR	Low-density lipoprotein receptor (Fragment)	−3.3732	0.0065
TMX1	Thioredoxin-related transmembrane protein 1	−3.3412	0.0015
PSMD4	26S proteasome non-ATPase regulatory subunit 4	−3.3016	0.0123
DNAAF5	Dynein axonemal assembly factor 5	−3.0889	0.0065
TST	Thiosulfate sulfurtransferase	−3.0300	0.0075
PXDN	Peroxidasin homolog	−3.0016	0.0003
OGFR	Opioid growth factor receptor	−2.9898	0.0177
ENG	Endoglin	−2.7407	0.0145
FAF1	FAS-associated factor 1	−2.5783	0.0183
MCM5	DNA replication licensing factor MCM5	−2.5649	0.0022
ADSS2	Adenylosuccinate synthetase isozyme 2	−2.4898	0.0118
GIPC1	PDZ domain-containing protein GIPC1	−2.4031	0.0494
ACO1	Cytoplasmic aconitate hydratase	−2.3932	0.0013
PRPSAP2	Phosphoribosyl pyrophosphate synthase-associated protein 2	−2.3817	0.0099
DNAJA3	DnaJ homolog subfamily A member 3, mitochondrial	−2.3163	0.0022
DLAT	Dihydrolipoyllysine-residue acetyltransferase component of pyruvate dehydrogenase complex, mitochondrial	−2.3145	0.0009
SNX5	Sorting nexin-5	−2.2927	0.0082
PCK2	Phosphoenolpyruvate carboxykinase [GTP], mitochondrial	−2.2247	0.0184
IGFBP7	Insulin-like growth factor-binding protein 7	−2.2145	0.0077
NDRG3	N-myc downstream-regulated gene 3 protein	−2.1762	0.0005
KRT73	Keratin, type II cytoskeletal 73	−2.1747	0.0458
PC	Pyruvate carboxylase, mitochondrial	−2.1747	0.0187
CORO7	Coronin	−2.1529	0.0035
POLDIP2	Polymerase delta-interacting protein 2	−2.1529	0.0011
ACOT2	Acyl-coenzyme A thioesterase 2, mitochondrial	−2.1328	0.0021
MAP2K2	Dual-specificity mitogen-activated protein kinase kinase 2	−2.1182	0.0009
MAGED2	Melanoma-associated antigen D2	−2.1069	0.0476
IDH3A	Isocitrate dehydrogenase [NAD] subunit alpha, mitochondrial	−2.0820	0.0012
ALDH2	Aldehyde dehydrogenase, mitochondrial	−2.0759	0.0003
SLC9A3R1	Na(+)/H(+) exchange regulatory cofactor NHE-RF1	−2.0756	0.0071

**Table 2 ijms-24-07531-t002:** Differentially expressed proteins in CuHL2-treated MDA-MB-231 cells.

Gene	Protein	Fold-Change	*p*-Value
HSPA6	Heat shock 70 kDa protein 6	135.1126	0.0001
HSPA1B	Heat shock 70 kDa protein 1B	12.4443	0.0000
SRXN1	Sulfiredoxin-1	4.7883	0.0157
BAG3	BAG family molecular chaperone regulator 3	4.4887	0.0045
G3BP2	Ras GTPase-activating protein-binding protein 2	3.6714	0.0310
THBS1	Thrombospondin-1	3.5660	0.0120
CHORDC1	Cysteine and histidine-rich domain-containing protein 1	3.5336	0.0057
FXR2	Fragile X mental retardation syndrome-related protein 2	3.4133	0.0198
SLC25A3	Phosphate carrier protein, mitochondrial	3.2324	0.0178
DNAJB1	DnaJ homolog subfamily B member 1	3.0959	0.0000
SLC25A5	ADP/ATP translocase 2	3.0855	0.0253
GCLM	Glutamate–cysteine ligase regulatory subunit	2.8046	0.0024
ATAD3A	ATPase family AAA domain-containing protein 3A	2.7501	0.0031
MMP1	Interstitial collagenase	2.7465	0.0051
MAP1LC3B2	Microtubule-associated proteins 1A/1B light chain 3 beta 2	2.7147	0.0338
HSPH1	Heat shock protein 105 kDa	2.7037	0.0038
SDHA	Succinate dehydrogenase [ubiquinone] flavoprotein subunit, mitochondrial	2.2722	0.0472
DNAJA1	DnaJ homolog subfamily A member 1	2.2136	0.0003
ZNF622	Zinc finger protein 622	2.1139	0.0070
AKAP2	A-kinase anchor protein 2	2.0990	0.0117
TMCO1	Calcium-load-activated calcium channel	2.0869	0.0164
SLC3A2	4F2 cell-surface antigen heavy chain	2.0386	0.0031
AHSA1	Activator of 90 kDa heat shock protein ATPase homolog 1	1.9963	0.0228
SMCHD1	Structural maintenance of chromosomes flexible hinge domain-containing protein 1	−8.2820	0.0014
NFKB2	Nuclear factor NF-kappa-B p100 subunit	−5.8093	0.0379
DPYD	Dihydropyrimidine dehydrogenase [NADP(+)]	−5.6316	0.0001
RRM2	Ribonucleoside-diphosphate reductase subunit M2	−5.2034	0.0018
FADS2	Acyl-CoA 6-desaturase	−5.0562	0.0169
DNAAF5	Dynein axonemal assembly factor 5	−4.3873	0.0202
U2AF1	Splicing factor U2AF 35 kDa subunit	−4.2363	0.0027
SDC4	Syndecan-4	−4.2206	0.0232
ACSL4	Long-chain-fatty-acid--CoA ligase 4	−3.7610	0.0026
CYP51A1	Lanosterol 14-alpha demethylase	−3.6858	0.0032
TP53	Cellular tumor antigen p53	−3.6284	0.0005
GFM1	Elongation factor G, mitochondrial	−3.4629	0.0032
LDLR	Low-density lipoprotein receptor	−3.2341	0.0284
OGFR	Opioid growth factor receptor OS = Homo sapiens	−3.2299	0.0139
RIN1	Ras and Rab interactor 1	−2.7981	0.0113
FAF1	FAS-associated factor 1	−2.6790	0.0022
MCM2	DNA replication licensing factor MCM2	−2.6450	0.0239
PCK2	Phosphoenolpyruvate carboxykinase [GTP], mitochondrial	−2.6115	0.0004
SUMF2	Inactive C-alpha-formylglycine-generating enzyme 2	−2.5908	0.0432
HNRNPUL1	Heterogeneous nuclear ribonucleoprotein U-like protein 1	−2.5669	0.0440
PXDN	Peroxidasin homolog	−2.5552	0.0031
MCM3	DNA replication licensing factor MCM3	−2.5299	0.0332
DNAJA3	DnaJ homolog subfamily A member 3, mitochondrial	−2.5193	0.0231
PACS1	Phosphofurin acidic cluster sorting protein 1	−2.4865	0.0405
POGLUT3	Protein O-glucosyltransferase 3	−2.3599	0.0119
SCYL1	N-terminal kinase-like protein	−2.3123	0.0020
FASN	Fatty acid synthase	−2.2785	0.0382
VIM	Vimentin	−2.2682	0.0343
MCM5	DNA replication licensing factor MCM5	−2.2670	0.0021
TRIM28	Transcription intermediary factor 1-beta	−2.2658	0.0431
CDC37	Hsp90 co-chaperone Cdc37	−2.2547	0.0236
YARS2	Tyrosine--tRNA ligase, mitochondrial	−2.2501	0.0139
SMC2	Structural maintenance of chromosomes protein 2	−2.2447	0.0288
ITGB4	Integrin beta-4	−2.2163	0.0362
PTK7	Inactive tyrosine-protein kinase 7	−2.1469	0.0128
CORO7	Coronin	−2.1154	0.0290
ACOT2	Acyl-coenzyme A thioesterase 2, mitochondrial	−2.1016	0.0002
TBC1D9B	TBC1 domain family member 9B	−2.0811	0.0103
MYO18A	Unconventional myosin-XVIIIa	−2.0581	0.0201
CALM2	Calmodulin-2	−2.0396	0.0306

A comparative analysis of differentially expressed proteins between treatments showed 32 proteins in common. Among down-regulated proteins, the treatments with CuHL1 and CuHL2 shared 18 proteins: TP53, MCM5, PCK2, CORO7, ACOT2, OGFR, FAF1, LDLR, NFKB2, DPYD, DNAAF4, PXDN, RRM2, FADS2, CYP51A1, SMCHD1, SDC4, and DNAJA3. Meanwhile, among up-regulated proteins, the complexes shared 14 proteins: HSPA1B, HSPH1, DNAJB1, DNAJA1, CHORDC1, BAG3, ZNF622, THBS1, MAP1LC3B2, SLC25A3, GCLM, MMP1, SRXN1, and SLC25A5.

**Table 3 ijms-24-07531-t003:** Ingenuity canonical pathways associated with the differentially expressed proteins in CuHL1-treated cells.

Canonical Pathways	*p*-Value	Proteins
EIF2 Signaling	4.78 × 10^−6^	ACTC1, MAP2K2, RPL10, RPL12, RPL21, RPL7A, RPS8
Unfolded protein response	6.97 × 10^−6^	DNAJA1, DNAJA3, DNAJB1, HSPA1A/HSPA1B, HSPH1
Induction of Apoptosis by HIV1	4.09 × 10^−5^	NFKB2, SLC25A3, SLC25A5, TP53
NRF2-mediated Oxidative Stress Response	7.13 × 10^−5^	ACTC1, DNAJA1, DNAJA3, DNAJB1, GCLM, MAP2K2
BAG2 Signaling Pathway	1.12 × 10^−4^	HSPA1A/HSPA1B, NFKB2, PSMD4, TP53
Aldosterone Signaling in Epithelial Cells	1.39 × 10^−4^	DNAJA1, DNAJB1, HSPA1A/HSPA1B, HSPH1, MAP2K2
Bladder Cancer Signaling	3.87 × 10^−4^	MAP2K2, MMP1, THBS1, TP53
Cell Cycle Control of Chromosomal Replication	6.08 × 10^−4^	MCM5, POLD1, TOP2A
Protein Ubiquitination Pathway	1.30 × 10^−3^	DNAJA1, DNAJB1, HSPA1A/HSPA1B, HSPH1, PSMD4
Ribonucleotide Reductase Signaling Pathway	1.61 × 10^−3^	NFKB2, RRM2, THBS1, TP53
HIF1α Signaling	3.34 × 10^−3^	HSPA1A/HSPA1B, MAP2K2, MMP1, TP53
Apoptosis Signaling	3.61 × 10^−3^	MAP2K2, NFKB2, TP53
Autophagy	3.82 × 10^−3^	MAP1LC3B2, MAP2K2, NFKB2, TP53
HER-2 Signaling in Breast Cancer	4.56 × 10^−3^	ARF3, MAP2K2, NFKB2, TP53

**Table 4 ijms-24-07531-t004:** Ingenuity canonical pathways associated with the differentially expressed proteins in CuHL2-treated cells.

Canonical Pathways	*p*-Value	Proteins
Unfolded protein response	1.28 × 10^−7^	DNAJA1, DNAJA3, DNAJB1, HSPA6, HSPA1A/HSPA1B, HSPH1
Sirtuin Signaling Pathway	1.10 × 10^−5^	MAP1LC3B2, NFKB2, PCK2, SDHA, SLC25A5, TP53, TRIM28
Induction of Apoptosis by HIV1	2.51 × 10^−5^	NFKB2, SLC25A3, SLC25A5, TP53
BAG2 Signaling Pathway	6.90 × 10^−5^	HSPA6, HSPA1A/HSPA1B, NFKB2, TP53
Aldosterone Signaling in Epithelial Cells	7.73 × 10^−5^	DNAJA1, DNAJB1, HSPA6, HSPA1A/HSPA1B, HASPH1
HIF1α Signaling	2.10 × 10^−4^	HSPA6, HSPA1A/HSPA1B, MMP1, TP53, VIM
Autophagy	2.50 × 10^−4^	CALM1, MAP1LC3B2, NFKB2, SLC3A2, TP53
LXR/RXR Activation	3.01 × 10^−4^	CYP51A1, FASN, LDLR, NFKB2
Cell Cycle Control of Chromosomal Replication	4.24 × 10^−4^	MCM2, MCM3, MCM5
Ribonucleotide Reductase Signaling Pathway	1.02 × 10^−3^	NFKB2, RRM2, THBS1, TP53
Immunogenic Cell Death Signaling Pathway	1.69 × 10^−3^	HSPA6, HSPA1A/HSPA1B, NFKB2
PI3K/AKT Signaling	1.85 × 10^−3^	CDC37, ITGB4, NFKB2, TP53
NRF2-mediated Oxidative Stress Response	3.41 × 10^−3^	DNAJA1, DNAJA3, DNAJB1, GCLM
Inhibition of Angiogenesis by TSP1	3.55 × 10^−3^	THBS1, TP53
MYC-Mediated Apoptosis Signaling	7.56 × 10^−3^	NFKB2, TP53
FAT10 Cancer Signaling Pathway	7.56 × 10^−3^	NFKB2, TP53

## Data Availability

The data presented in this study are available in the article.

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
