# Peer review of "Finding New Molecular Targets of Two Copper(II)-Hydrazone Complexes on Triple-Negative Breast Cancer Cells Using Mass-Spectrometry-Based Quantitative Proteomics"

_ijms, 2023, doi:10.3390/ijms24087531_

Round 1

Reviewer 1 Report

It is interesting to study how copper complexes manage to interfere in cellular proteomics. I believe that this study is presented as an interesting approach to the mechanism of action of possible metallo drugs. In this sense, I believe that this work is a solid contribution to improving the manuscripts to be published in the future. Since from the bioinorganic field, the effect that metallo complexes cause at the protein level in cells is little boarded in the literature. I strongly suggest that this manuscript should be published after some minor corrections. 

1.- Please, improve the quality and pixel sizes of Figures 1, 2, 3, and 4. Is extremely necessary.

2.- Please, add to Figure 4, the information used to generate the interactions protein-protein depicted in the figure, and the version of STRING used to construct it.

Author Response

It is interesting to study how copper complexes manage to interfere in cellular proteomics. I believe that this study is presented as an interesting approach to the mechanism of action of possible metallo drugs. In this sense, I believe that this work is a solid contribution to improving the manuscripts to be published in the future. Since from the bioinorganic field, the effect that metallo complexes cause at the protein level in cells is little boarded in the literature. I strongly suggest that this manuscript should be published after some minor corrections. 

1.- Please, improve the quality and pixel sizes of Figures 1, 2, 3, and 4. Is extremely necessary.

This request was addressed.

2.- Please, add to Figure 4, the information used to generate the interactions protein-protein depicted in the figure, and the version of STRING used to construct it.

According to the reviewer´s suggestion we have been added the information in Figure 4.

Reviewer 2 Report

Dear Authors,

In my opinion, the manuscript “Finding new molecular targets of two copper(II)-hydrazones complexes on triple negative breast cancer cells using Mass Spectrometry-based Quantitative Proteomics” was written by Lucia Balsa and co-authored, described the mechanisms of hydrazone copper complexes which exert their antitumoral effect in TNBC cells. Understanding the mechanisms of anticancer action of new metallocomplex compounds is a current and contemporary research topic. The interesting finding is the regulation of proteins related to lipid synthesis and metabolism that could lead decrease in lipid levels. The background in the introduction is well-written. Quantitative proteomics and functional bioinformatics methods were used to present molecular mechanisms of action. However, the presentation of results require better quality and the article needs some correction according to the comments below: 

1.      The charge of CuHL1 and CuHL2 is ‘+’ and these name of complex forms are misleading. It is good idea to distinguish ligands, but maybe it will be better to write ‘L1’ and ‘L2’ without the upper index.

2.      Please explain why the coordination number for copper ion in CuHL1 is 4 and for CuHL2 is 5? Where is this knowledge from? What research has it proven? Why CuHL1 has only one water molecule bound in Cu and CuHL2 has two molecules of water? If you obtain this in previous papers please give references in the caption Figure 1.

3.      In Table 1 the value of decimal fractions should be written with a dot ‘.’ Not decimal ‘,’. Please revise it in whole your work.

4.      The sidebar charts in figure 3 are completely unreadable. Please put better resolution figure or put these charts separated and enlarge.

5.      Due to the very poor quality of figure 4, the interpretation and understanding of the PPI analysis is practically impossible. A better quality drawing with a larger font is required.

6.      List with explanations of all abbreviations used should be provided at the end of the manuscript.

7.      In ‘Author Contributions’ there is no information about one co-author Verónica Ferraresi-Curotto, please add a contribution from this author. I hope this is just a minor oversight.

8.      Please read carefully the whole article with attention to editorial and typos bags. In my opinion, there are small and do not diminish work, but should be corrected, e.g. too many spaces, too many dots ect.

Author Response

Dear Authors,

In my opinion, the manuscript “Finding new molecular targets of two copper(II)-hydrazones complexes on triple negative breast cancer cells using Mass Spectrometry-based Quantitative Proteomics” was written by Lucia Balsa and co-authored, described the mechanisms of hydrazone copper complexes which exert their antitumoral effect in TNBC cells. Understanding the mechanisms of anticancer action of new metallocomplex compounds is a current and contemporary research topic. The interesting finding is the regulation of proteins related to lipid synthesis and metabolism that could lead decrease in lipid levels. The background in the introduction is well-written. Quantitative proteomics and functional bioinformatics methods were used to present molecular mechanisms of action. However, the presentation of results require better quality and the article needs some correction according to the comments below: 

  1. The charge of CuHL1and CuHL2 is ‘+’ and these name of complex forms are misleading. It is good idea to distinguish ligands, but maybe it will be better to write ‘L1’ and ‘L2’ without the upper index.

 According to the reviewer´s suggestion, the ligands are referred as L1 and L2 in the corrected version. The abbreviated formula of the coordination compounds are included in the Introduction and the ligands are named in the Results and discussion section.

  1. Please explain why the coordination number for copper ion in CuHL1is 4 and for CuHL2 is 5? Where is this knowledge from? What research has it proven? Why CuHL1 has only one water molecule bound in Cu and CuHL2 has two molecules of water? If you obtain this in previous papers please give references in the caption Figure 1.

The coordination of the copper center in each compound is obtained from the cristal structure solved by XRD in monocrystalline samples, previously reported. This information is discussed in references 19 and 21. For clarity, a sentence is added in the text and the references are included in the caption of figure 1, as suggested.

Because CuHL1 complex crystallizes as a monohydrate, [Cu(HL1)(H2O)](NO3).H2O, a new figure, containing the hydration water molecule, is included in the corrected version of the manuscript, for clarity.

  1. In Table 1 the value of decimal fractions should be written with a dot ‘.’ Not decimal ‘,’. Please revise it in whole your work.

Done

  1. The sidebar charts in figure 3 are completely unreadable. Please put better resolution figure or put these charts separated and enlarge.

Done

  1. Due to the very poor quality of figure 4, the interpretation and understanding of the PPI analysis is practically impossible. A better quality drawing with a larger font is required.

According to the reviewer´s suggestion, we have added a better quality drawing in Figure 4 in the revised versión of manuscript.

  1. List with explanations of all abbreviations used should be provided at the end of the manuscript.

According to the reviewer´s suggestion, we have added a list of abbreviation in the revised versión of manuscript.

  1. In ‘Author Contributions’ there is no information about one co-author Verónica Ferraresi-Curotto, please add a contribution from this author. I hope this is just a minor oversight.

Done

  1. Please read carefully the whole article with attention to editorial and typos bags. In my opinion, there are small and do not diminish work, but should be corrected, e.g. too many spaces, too many dots ect.

According to the reviewer´s comment, we have checked and corrected typos in the revised versión of manuscript.

Reviewer 3 Report

In this work, the authors performed quantitative proteomic analysis of MDA-MB-231cells treated with copper(II)-hydrazone complexes to reveal novel molecular mechanisms behind the antitumor effects of these two complexes in TNBC cells. The results revealed that both CuHL1 and CuHL2 treatment up-regulated the proteins involved in ER stress and UPR and down-regulated GOF-mutant p53 and proteins related to lipid synthesis and metabolism. This provides some novel insights into the anticancer effects of copper metallodrugs. However, there are some issues/concerns with the manuscript.

Issues:

In Results and Discussion part, under the subtitle “2.2. Label-Free Mass Spectroscopy quantification of proteins isolated from MDA-MB-231 cells following treatment with CuHL1 and CuHL2”, how many proteins in total were identified in CuHL1- and CuHL2- treated cells? If the threshold of the fold change is relaxed to 1.5, how many proteins were identified to be differentially expressed under the two conditions?

Figure 2: How many replicates were done for the data in the figure? To obtain reliable results, the data should be from at least three individual replicates.

Figure 3: Typically, gene oncology analysis includes three categories: Biological Process, Molecular Function, and Cell Component. Why do some of the panels only include two categories? Please add “Cell” before “Component” in panels c and d.

Table 1 & 2: Did authors compare whether there are some common differentially expressed proteins between the two copper complex treatments?  This could help to figure out the shared features of the two copper complexes.

Quantitative proteomics was employed to investigate the influences on protein expression of the two copper complexes in MDA-MB-231 cells. Did the authors perform proteomics analysis in some other breast cancer cell lines, like MCF7 as well as parallel normal cell lines to rule out the non-specific change following the treatment of the copper complexes in TNBC cells?

Author Response

In Results and Discussion part, under the subtitle “2.2. Label-Free Mass Spectroscopy quantification of proteins isolated from MDA-MB-231 cells following treatment with CuHL1 and CuHL2”, how many proteins in total were identified in CuHL1- and CuHL2- treated cells? If the threshold of the fold change is relaxed to 1.5, how many proteins were identified to be differentially expressed under the two conditions?

A total of 1656 and 1659 proteins were identified for with CuHL1 and CuHL2-treated cells respectively. Using the recommended threshold (2), a total of 69 proteins (28 proteins up-regulated and 41 proteins down-regulated) were identified for CuHL1-treated cells whilst a total of 63 proteins (23 proteins up-regulated and 40 proteins down-regulated) were identified to be differentially expressed for CuHL2-treated cells. However, if the threshold of the fold change is relaxed to 1.5, a total of 98 proteins (68 up-regulated and 130 down-regulated) were identified for CuHL1 treatment whilst a total of 143 proteins (39 up-regulated and 104 down-regulated) were discovered.

Figure 2: How many replicates were done for the data in the figure? To obtain reliable results, the data should be from at least three individual replicates.

The data should be from the three individual replicates.

Figure 3: Typically, gene oncology analysis includes three categories: Biological Process, Molecular Function, and Cell Component. Why do some of the panels only include two categories? Please add “Cell” before “Component” in panels c and d.

CuHL1 did not show enriched terms of Molecular Functions for down-regulated proteins; whilst, CuHL2 did not show enriched terms of Cell Component for up-regulated proteins. This request was addressed.

Table 1 & 2: Did authors compare whether there are some common differentially expressed proteins between the two copper complex treatments?  This could help to figure out the shared features of the two copper complexes.

According to the reviewer´s comment, we have included a comparative analysis for common differentially expressed proteins between the two copper complexes in the revised versión of manuscript. Comparition of differentially expressed proteins between treatments showed 32 proteins in common. Among down-regulated proteins, CuHL1 and CuHL2 shared 18 proteins: TP53, MCM5, PCK2, CORO7, ACOT2, OGFR, FAF1, LDLR, NFKB2, DPYD, DNAAF4, PXDN, RRM2, FADS2, CYP51A1, SMCHD1, SDC4 and DNAJA3. Meanwhile, among up-regulted proteins, the complexes shared 14 proteins: HSPA1B, HSPH1, DNAJB1, DNAJA1, CHORDC1, BAG3, ZNF622, THBS1, MAP1LC3B2, SLC25A3, GCLM, MMP1, SRXN1 and SLC25A5.

Quantitative proteomics was employed to investigate the influences on protein expression of the two copper complexes in MDA-MB-231 cells. Did the authors perform proteomics analysis in some other breast cancer cell lines, like MCF7 as well as parallel normal cell lines to rule out the non-specific change following the treatment of the copper complexes in TNBC cells?

Thanks to reviewer for comment. The main goal of this work is to explore the principal targets related to anticancer activity of copper complexes on TNBC. Currently, this kind of tumor has a low efficacy treatment since it produces important adverse effects and a high rate of metastatic recurrences so we focus our work to develop new strategies using therapeutic agents to improve and optimize the treatment of TNBC. Anyway, in future works it would be interesting include these comparative studies using positive-hormone breast cancer cells (MCF7) and non-tumoral cells.

Round 2

Reviewer 2 Report

After the introduced changes, the quality of the article gained importance. Recommends the manuscript for publication in IJMS in the current form. 

Author Response

Thanks the reviewer for your comments

Reviewer 3 Report

This revised manuscript well addressed my previous comments. Overall, the work showed the a good comparison of the proteomics results from the two copper(II)-hydrazone complexes, CuHL1 and CuHL2. Afterwards, authors’ analyses of Gene Ontology, protein-protein interaction, and ingenuity canonical pathways provide a better understanding of the mechanism behind the anticancer effects of the copper complexes. Below are some minor issues/concerns with the manuscript.

Issues:

Authors discovered 28 up- and 41 down-regulated proteins in CuHL1-treated cells and 23 up- and 40 down-regulated proteins in CuHL2-treated cells. In total, 14 upregulated and 18 downregulated proteins were shared by the treatments of the two metal complexes. It is not surprising that there were more common terms with GO of analysis of upregulated proteins compared to that of down regulated proteins, as the proportion of common upregulated proteins (50% and 61%) was greater in both treatments. Authors should consider including discussion like for results interpretation.

For the protein-protein interaction analysis, authors mapped the PPI network with the proteins upregulated and downregulated under CuHL1 and CuHL2 treatments, providing a good comparison between the two treatments. But some information in the graphs is a little bit redundant. Maybe mapping the PPI network with the common proteins from the both treatments could help show the results more clearly and draw conclusions easily.

Author Response

Thanks of reviewer for the comment.

According to the reviewer´s suggestion, we have added the PPI network with common proteins between CuHL1 and CuHL2 treatments (see new Fig 6 in the revised version of the manuscript). Besides, we have added a paragraph with a discussion about this.